# SurvPFN: Towards Foundation Models for Survival Predictions

**Samuel Böhm** [1]   **Lennart Purucker** [2 3]   **Frank Hutter** [2 4 3]   **Pascal Schlosser** [1 5 6]

## Abstract

Tabular foundation models (TFMs) have made rapid progress in standard classification and regression, but time-to-event survival prediction tasks have remained largely untouched. Unlike in standard regression tasks, survival prediction models must account for censored data. Standard TFMs cannot handle natively censored data, leading to biased and inaccurate predictions, making them unsuitable for real-world applications. To overcome this fundamental limitation, we propose `SurvPFN`, a prior-data fitted network (PFN), for survival prediction tasks. We pretrain `SurvPFN` on millions of synthetic survival prediction tasks to learn survival via distributional regression that accounts for censored data. `SurvPFN` works by (1) generating data with Weibull event times and a non-informative censoring mechanism; (2) integrating a censored event indicator; and (3) minimizing a censored negative log-likelihood. On SurvSet, a collection of real-world survival tasks, `SurvPFN` is competitive with classical and deep survival baselines without per-dataset fitting, a survival-specific architecture, or feature engineering. We show that survival can be treated as a continuous-time distributional regression problem with censored loss, unlocking the power of PFNs for time-to-event predictions. Our code is available at: https://github.com/genepi-freiburg/SurvPFN.

---

[1]Institute of Epidemiology and Prevention, Medical Center – University of Freiburg, Freiburg, Germany [2]PriorLabs, Germany [3]Department of Computer Science, University of Freiburg, Germany [4]ELLIS Institute Tübingen, Germany [5]Department of Epidemiology, Johns Hopkins Bloomberg School of Public Health, Baltimore, Maryland, US [6]CIBSS - Centre for Integrative Biological Signalling Studies, University of Freiburg, Freiburg, Germany. Correspondence to: Samuel Böhm <samuel.boehm@uniklinik-freiburg.de>, Pascal Schlosser <pascal.schlosser@uniklinik-freiburg.de>.

*Proceedings of the $2^{nd}$ ICML Workshop on Foundation Models for Structured Data*, Seoul, South Korea. 2026. Copyright 2026 by the author(s).

## 1. Introduction

The introduction of prior-data fitted networks for tabular data (Hollmann et al., 2023), with `TabPFN` and its successors, sparked a wave of research on tabular foundation models (Dooley et al., 2023; Qu et al., 2025; Eremeev et al., 2026; Küken et al., 2026). PFNs are pretrained once on synthetic datasets and then solve new tasks through approximate Bayesian inference, without fitting per data set (Hollmann et al., 2023; 2025; Qu et al., 2026). Progress has concentrated on classification and regression, with prior generators, training recipes, and benchmarks all scaling up accordingly. Survival analysis, despite being a classical tabular task with broad applications in medicine, economics, and reliability engineering, has remained largely outside this wave until very recent concurrent work (Kim et al., 2026; Seletkov et al., 2026).

A particular challenge in survival data is censoring. In some cases the event of interest (e.g., death, failure, cure) is not observed during the follow-up period. This leads to the most common form of censoring, *right censoring*, where the event has not occurred by the end of the study period. For right-censored instances, the recorded time is a lower bound on the true event time. Naively treating these as regression targets, or dropping them, would introduce a systematic bias. Moreover, censoring is not simply missing data: it provides partial information about the event time, which is informative for the task. Existing survival methods such as the Kaplan–Meier estimator (Kaplan & Meier, 1958), the Cox proportional hazards model (Cox, 1972), and modern deep learning approaches (Katzman et al., 2018) explicitly model censoring in their likelihood functions or loss terms, ensuring unbiased estimation.

We ask wheter an PFN can learn survival as a distributional-regression task. For this, we turned to `NanoTabPFN`, a small, computationally efficient TFM from the `TFM-Playground` (Pfefferle et al., 2025), and adapted it for survival prediction. We designed a synthetic survival task that mimics real-world conditions. Using structural causal models (SCMs), we generated event times from a Weibull distribution, a common choice in survival analysis, and introduced non-informative censoring to simulate realistic scenarios. The binary censoring information was provided as an additional feature, ensuring the model learns to account for uncertainty. We then minimize the

censored negative log likelihood, a standard loss function for survival tasks that naturally handles censored data. We evaluate `SurvPFN` on 22 real-world datasets from SurvSet, spanning clinical, economic, and reliability domains, and compare against a representative slice of the survival literature: *Cox proportional hazards* (Cox, 1972) as the dominant classical baseline, *Random Survival Forests* (Ishwaran et al., 2008) as the standard non-parametric ensemble, *DeepSurv* as the most widely used deep-learning approach, and a recent tabular-foundation-model alternative we refer to as *BinSurv* (Kim et al., 2026). Across this comparison, our model is competitive with all four — recovering the same performance band by directly regressing to continuous time, with no per-dataset tuning, no survival-specific architecture and no feature engineering.

## 2. Related Work

**Prior-data Fitted Networks** PFNs use synthetic data to train a model that adapts to new tasks in a single forward pass, mimicking Bayesian predictions without per-task gradient updates (Hollmann et al., 2025; Qu et al., 2026). State-of-the-art models such as `TabPFN-2.6` (Grinsztajn et al., 2025) and `TabICLV2` (Qu et al., 2026) are trained on millions of datasets with advanced training techniques. Competitive results can nevertheless be achieved in a small-data regime with little compute: `NanoTabPFN` and the TFM playground provide a small, fast-to-train testbed for iterating on priors and training objectives (Pfefferle et al., 2025).

The choice of prior is not incidental: it directly shapes downstream performance (Hollmann et al., 2023), and allows PFN designers to encode inductive bias. We build on the publicly released `TabICL` prior (Qu et al., 2025), a robust baseline based on SCMs with a selection of activation functions on the nodes. Specifically, we use the smaller variant released by Reuter and Robertson (Robertson et al., 2025; Reuter et al., 2026).

`SurvPFN` builds directly on the `TabPFN` architecture in its small, user-friendly `NanoTabPFN` reimplementation (Pfefferle et al., 2025), with minimal changes targeted at categorical features. It is trained on an SCM based prior. Event times are drawn from a Weibull distribution under both proportional and non-proportional hazard assumptions, with non-informative censoring applied on top.

**Survival Analysis** The classical approach for survival analysis is the Cox proportional hazards model (CoxPH) (Cox, 1972), a semi-parametric model that assumes each subject's hazard is a fixed multiple of a shared baseline hazard. CoxPH is cheap to fit and straightforward to interpret, but its proportional-hazards assumption is often violated in practice, and it cannot capture nonlinear covariate effects. Random Survival Forests (RSF) (Ishwaran et al., 2008) ad-

dress both limitations through a nonparametric ensemble that handles censoring natively during tree splitting. Deep-Surv (Katzman et al., 2018) replaces the linear predictor in the Cox model with a neural network, allowing complex nonlinear covariate effects while retaining the proportional-hazards constraint.

Two concurrent works adapt tabular foundation models to survival. `BinSurv` (Kim et al., 2026) reformulates survival as a sequence of binary classifications: time is discretized into $K$ bins, and each subject is represented by up to $K-1$ tuples, with labels beyond the censoring time dropped. This lets an off-the-shelf TFM *classifier* (e.g. `MITRA`, `TabPFN-2.5` (Grinsztajn et al., 2025; Zhang et al., 2025)) perform survival analysis, at the cost of a resolution–computation tradeoff: more bins improve temporal resolution but lengthen the context, so larger datasets must be subsampled. A recent preprint, Survival In-Context (`SIC`) (Seletkov et al., 2026), takes a different route: it continues pretraining `TabICL` on a dedicated survival prior, built from SCMs, and attaches a DeepHit-style discrete head (Lee et al., 2018) trained with a ranking-weighted loss. `SIC` reports promising results on a curated set of clinical datasets, but at the time of writing, neither code nor model checkpoints are publicly available, and the published results report only the C-index, which precludes inclusion of `SIC` in our empirical comparison reported below.

In contrast, we train a small regression PFN from scratch on a survival prior, model time continuously through a fine-grained density, and let the event indicator enter as a feature column without further feature engineering.

## 3. SurvPFN

In the following we describe how we generate our survival specific prior. We then describe our adjustments to the `NanoTabPFN` model, how we train it, and how its survival event-time predictions work.

### 3.1. Prior Generation

We sample synthetic datasets from structural causal models (SCMs), following the `TabICL` prior (Qu et al., 2025) as implemented by Reuter and Robertson (Robertson et al., 2025; Reuter et al., 2026), with raised categorical sampling frequency and cardinality. Each dataset has $d \in \{2, \ldots, 10\}$ covariates and up to 1000 rows.

**Covariates and risk score:** we sample an SCM and propagate it to per-subject node values. The final topological node, standardised and scaled, is the risk score $\eta_i$; the remaining nodes (a subset retaining at least one ancestor of $\eta_i$) form the covariates $x_i$.

**Event times:** we use the Weibull family because it allows for diverse, increasing and decreasing baseline hazards while its closed form makes inverse-CDF sampling

cheap. Per dataset we draw a shape $k > 0$ and rate $\lambda > 0$. Event times are sampled by inverse-CDF from $U_i \sim \mathrm{Uniform}(0, 1)$:

$$T_i = \tfrac{1}{\lambda} \left( -\log U_i \right)^{1/k_i} \exp\left( -\eta_i/k_i \right), \qquad (1)$$

so $\eta_i$ acts as a linear predictor on the log-hazard. With a 50% chance we break proportionality and yield another node to calculate per-subject hazard shapes, which induces crossing survival curves (Appendix B.2.1).
**Censoring:** we apply non-informative administrative and loss-to-follow-up censoring, yielding observed times $t_i = \min(T_i, C_i)$ and indicators $\delta_i = \mathbf{1}[T_i \leq C_i]$ (Appendix B.2.2).

### 3.2. Model

We use `NanoTabPFN` regressor from the `TFMplayground` library as our backbone, with one addition: a two-routed target encoder that processes observed events and censored rows through separate encoders of identical architecture, allowing the model to learn distinct representations for true event times and lower bounds. Our model is trained for max. 1000 rows and $10 + 1$ feature columns. In comparison to large TFMs (`TabICL` 27.0M parameters, `TabPFN-2.5` 10.1M parameters, `MITRA` 72.0M parameters) `SurvPFN` is of smaller size (6 layers, 192 embedding dimensions, 4.4M parameters).

**Training objective** We train with an IPCW-weighted right-censored negative-log-likelihood loss: observed events contribute $-\frac{1}{\hat{G}(t_i)} \log \hat{f}(t_i \mid x_i)$, where $\hat{G}(t)$ is the Kaplan–Meier estimate of the censoring survival function fitted on the training context, and censored rows contribute $-\log \hat{S}(c_i \mid x_i)$, penalising the model for placing probability mass before the censoring time. We refer to this as *native* training (`SurvPFN [N]`), as it operates purely on observed data and transfers directly to real-world pretraining. We additionally add a differentiable pairwise ranking term to align predicted survival times with the underlying risk order; the combined objective is our headline (`SurvPFN [NR]`) variant. Two more variants that exploit oracle access during synthetic pretraining are described in Appendix B.3; all four perform comparably on SurvSet. We use default `NanoTabPFN` settings without hyperparameter tuning; baselines are tuned to be competitive (Appendix B.1).

**Survival prediction** At inference, the softmax output is read as a discrete density $\hat{p}_j$, from which $\hat{S}(t) = 1 - \sum_{j:\, t_j \leq t} \hat{p}_j$ and the predicted median $t^*$ with $\hat{S}(t^*) = 0.5$ follow directly.

### 3.3. Data Preprocessing

Let the context be $\mathcal{D} = \{(x_i, \delta_i, t_i)\}_{i=1}^n$, where $x_i \in \mathbb{R}^d$ are covariates, $\delta_i \in \{0, 1\}$ is the event indicator ($\delta_i = 1$ for an observed event, 0 for right-censoring), and $t_i$ is the observed time. We append the indicator to the covariates, $\tilde{x}_i = [x_i, \ \delta_i] \in \mathbb{R}^{d+1}$, and pair it with target $t_i$, so the model can tell which context rows are observed events and which are censored lower bounds. The event-time distribution for a query $x_q$ is then

$$\hat{p}(t \mid \tilde{x}_q, \mathcal{D}), \qquad \tilde{x}_q = [x_q, \ \delta_q{=}1]. \qquad (2)$$

The query indicator is clamped to $\delta_q = 1$, so the model is always asked for an event time rather than a censoring time.

**Features.** Continuous features are $z$-normalised independently per dataset on the training split; categorical features are passed directly to the model.
**Target.** The event-time axis is partitioned into 1000 equal-mass buckets, computed once from pooled synthetic training data; event times are first log1p-transformed, then $z$-normalised using the mean and standard deviation of observed (uncensored) events only.
**Ablation.** To isolate the effect of the indicator, we retrain each model with $\delta_i$ fixed to 1 across all rows, removing the model's ability to distinguish events from censored lower bounds in the context.

### 3.4. Experiments

We train our model exclusively on the synthetic prior and evaluate on real-world datasets from SurvSet (Drysdale, 2022), a curated collection of publicly available survival datasets. Matching specifications we selected SurvSet datasets with at most 1000 rows and 10 features, yielding a subset of 22 datasets. All models are evaluated under 5-fold cross-validation. For CoxPH, RSF, and DeepSurv, an inner split within each training fold is used for hyperparameter tuning. Ablations, with an uninformative event indicator feature, isolate how much the model relies on explicit censoring information.

**Metrics** Survival models are scored on three axes: Discrimination, Accuracy and Calibration (Lillelund et al., 2025). The weighted concordance index (C-index, higher is better) (Uno et al., 2011) measures discrimination. The Integrated Brier Score (IBS, lower is better) (Graf et al., 1999) measures prediction: the time-averaged squared error between the predicted survival function and the true event indicator. The Integrated Calibration index (ICI, lower is better) assesses the absolute difference between predicted survival probabilities and smoothed survival frequencies (Austin et al., 2020).

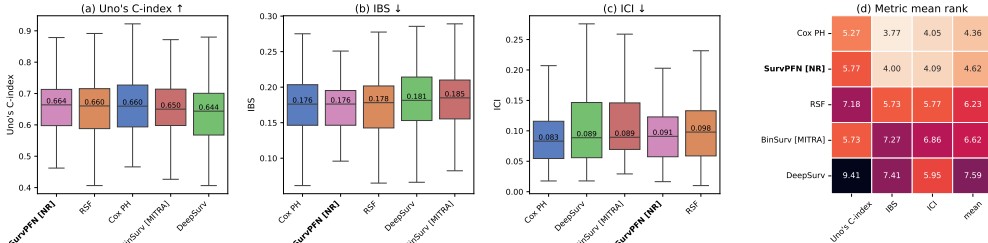

*Figure 1.* Performance across 22 SurvSet datasets under 5-fold cross-validation. Panels (a)–(c) show the distribution of Uno's C-index, Integrated Brier Score (IBS), and Integrated Calibration Index (ICI) across folds and datasets, with median values annotated; arrows indicate the direction of better performance; Outliers are not shown. Panel (d) shows the average per-metric rank of each model and its mean across the three metrics. Ranks are calculated across all n=12 models, full ranking results are shown in appendix (Table 2).

## 4. Results

### 4.1. Real-world performance on SurvSet

Our model is competitive across the board. No pairwise comparison between models reaches statistical significance, but on average `SurvPFN` matches or slightly exceeds RSF, DeepSurv, and `BinSurv`. CoxPH performs best in our evaluation scenario. Hence, a regression PFN, trained once from scratch on a synthetic Weibull prior, lands in the same performance band as task-specific survival models that are tuned per dataset – without per-task fitting and without a survival-specific architecture. Detailed per-dataset evaluations are shown in the Appendix Tables 1, 2.

**Ablation: removing the event indicator.** By forcing the event indicator to $\delta_i = 1$ across all rows, the model cannot tell events from censored observations. Discrimination is largely preserved, but calibration degrades noticeably, with the ICI rising (Table 1). The ranking signal in the data is recoverable without explicit censoring information, but locating $\hat{S}(t \mid x)$ at the right absolute level requires the model to know which context rows are lower bounds and which are observed events.

### 4.2. Discussion

A single inference on our small regression PFN, pretrained once on a generic Weibull-based synthetic prior, can match dataset-tuned classical and deep survival baselines, without per-dataset fitting. This zero-shot deployment can be an advatage especially in small cohorts: the model does not need per dataset training or hyperparameter tuning. Robustness at small $n$ is visible within our comparison: on the smallest cohorts (`Bergamaschi`, $n = 82$; `ovarian`, $n=26$; `glioma`, $n=37$) versions of SurvPFN mostly beat tuned baselines (Table 1). Performance also held up on datasets that fell outside the prior's typical shape: SurvSet contains fully categorical datasets and categorical variables with more levels than the prior generates, yet results on these sit in the same band as the rest of the benchmark (Table 1).

Our results indicate that the standard regression-PFN recipe is sufficient: censoring information together with a right censored loss is enough to capture survival as a lower bound problem.

**The event indicator carries information** Hiding the event indicator from the context affects model learning. While the C-index remains stable, IBS and ICI increase, as the model struggles to distinguish events from lower bounds in absolute time. This aligns with Kaplan–Meier intuition that relative risk ordering is recoverable from observed time alone, but absolute calibration requires the event signal.

**Limitations** We restrict evaluation to small datasets ($\leq$ 1000 rows, $\leq$ 10 features), a choice driven by the goal of fast iteration on the prior and training objective. Scaling to larger cohorts will require a larger backbone and a scaled prior. Further, the prior generates event times exclusively from a Weibull distribution. While we capture both PH and non-PH regimes, more expressive families may be needed to cover the full diversity of real-world survival curves. Finally, we model a single event of interest with static baseline covariates. Competing risks, where multiple event types preclude one another, and time-varying covariates, are both common in practice and not addressed here.

**Where to push next** Several directions follow naturally from the limitations. The backbone and operating regime: `NanoTabPFN` is deliberately small, and recent regression PFNs (Qu et al., 2026; Hollmann et al., 2025) suggests substantially stronger predictive densities from larger models on richer priors. Pushing into higher feature counts and larger context sizes is precisely where classical baselines like CoxPH start to struggle, and is where we would expect a foundation-model approach to overtake task-specific baselines rather than match them. The training signal: whether a right-censored NLL is the most informative objective is open, particularly because oracle access is available throughout pretraining and could be used for strong training signals.

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

# A. Acknowledgments

This research was funded by the Deutsche Forschungsgemeinschaft (DFG, German Research Foundation) – Project-ID 499552394 – SFB 1597, and Germany's Excellence Strategy (CIBSS – EXC-2189 – Project-ID 390939984).

# B. Appendix

## B.1. Training and Hyperparameter Optimization Details

**SurvPFN training** We use the default `NanoTabPFN` regressor architecture from the `TFMplayground` library. All four loss variants share the following training configuration: 200 epochs of 240 steps each, batch size 16 with gradient accumulation over 8 micro-batches (effective batch size 128). Each step samples a fresh synthetic dataset from the prior with 2–10 covariates and 100–1000 rows; the train fraction within each context is sampled uniformly from $[0.7, 0.8]$. We use the final-epoch model. Training and evaluation use one NVIDIA H100 GPU.

**Baseline hyperparameter optimization** For CoxPH, RSF, and DeepSurv, hyperparameters are selected by grid search on an $80/20$ stratified validation split of each training fold, maximising the C-index on the validation split. The selected configuration is then refit on the full training fold. The search spaces are:

- **CoxPH:** $\ell_2$ penalty $\alpha \in \{10^{-6}, 10^{-3}, 10^{-1}\}$. Categorical features are one-hot-encoded for the CoxPH model. Implemented with `lifelines`.

- **RSF:** number of trees $\in \{50, 100, 200\}$; minimum samples per split $\in \{2, 5, 10, 20\}$. Implemented with `scikit-survival`.

- **DeepSurv:** hidden layers $\in \{(128, 32), (256, 32), (512, 32)\}$; dropout $\in \{0.0, 0.1\}$; learning rate $\in \{10^{-3}, 10^{-4}, 10^{-5}\}$. Weight decay is fixed at $10^{-4}$, training runs up to 300 epochs with early stopping (patience 20). Re-implemented from original GitHub repository since code has broken dependencies.

**BinSurv** The original BinSurv paper does not identify a single optimal number of time bins $K$, and instead reports results averaged across a grid of discretizations. We follow the same protocol, evaluating with $K \in \{4, 5, 10, 15, 20\}$ and averaging metrics across the five runs. As the underlying classifier we use `MITRA` (Zhang et al., 2025), which achieved the best performance among the TFMs evaluated in the original paper.

## B.2. Prior Generation

### B.2.1. BREAKING PROPORTIONALITY

To break propotionality we calculate per subjects $k_i$ instead of fixed $k$ per dataset. This leads to non proportional hazards with different baselien shapes. To do so we yield another node from the SCM to get a standardised shape signal $s_i$. This is used to calculate per subject $k_i$.

$$k_i = \text{clip}\big(k \cdot \text{softplus}(s_i),\ 0.1,\ 10\big)$$

### B.2.2. CENSORING MECHANISM

Censoring is applied to the uncensored event times $T_i$ via two non-informative mechanisms.

**Administrative censoring** We set a dataset-level follow-up horizon at an event-time quantile, $\text{FU} = Q_T(q)$ with $q \sim \text{Uniform}(0.15, 0.95)$. With probability 0.05 enrolment is a hard global cutoff, $a_i = \text{FU}$ for all subjects. Otherwise subjects enter under staggered enrolment: a per-dataset window $w \sim \text{Beta}(1, 4)$ and front-loading parameter $\alpha \sim \text{Uniform}(0.3, 1.5)$ give per-subject entry times $e_i = b_i\, w\, \text{FU}$ with $b_i \sim \text{Beta}(\alpha, 1)$, so $a_i = \text{FU} - e_i$.

**Loss to follow-up** A per-dataset fraction $\rho \sim \text{Uniform}(0, 0.4)$ of subjects are additionally censored at a dropout time $r_i \sim \text{Uniform}(0, a_i)$.

The censoring time is $C_i = \min(a_i, r_i)$, and we observe $t_i = \min(T_i, C_i)$ with event indicator $\delta_i = \mathbf{1}[T_i \leq C_i]$.

## B.3. Loss Functions

We compare four training objectives.

**Native**   The right-censored negative log-likelihood operates purely on observed data, with inverse probability of censoring weighting (IPCW) applied to event terms to correct for informative censoring:

$$\mathcal{L}_{\text{native}} = -\frac{1}{N} \sum_{i=1}^{N} \left[ \delta_i \cdot \frac{1}{\hat{G}(t_i)} \log \hat{f}(t_i \mid x_i) + (1 - \delta_i) \log \hat{S}(t_i \mid x_i) \right], \tag{3}$$

where $\hat{G}(t) = \hat{P}(C > t)$ is the Kaplan–Meier estimate of the censoring survival function, fitted on the training split of each context. Events at times where few subjects remain at risk are upweighted by $1/\hat{G}(t_i)$; censored rows contribute the log-survival past the censoring time without reweighting.

**Oracle**   During synthetic pretraining the true event time $T_i$ is available for all subjects, including those that would be censored under the synthetic censoring mechanism. The oracle loss regresses to $T_i$ directly with a standard (non-censored) NLL:

$$\mathcal{L}_{\text{oracle}} = -\frac{1}{N} \sum_{i=1}^{N} \log \hat{f}(T_i \mid x_i), \tag{4}$$

This variant is not deployable on real data, where $T_i$ is not observed for censored subjects

**Ranking term**   On top of either base loss, we optionally add a differentiable pairwise ranking term that encourages the model's predicted mean event time to respect the oracle risk ordering induced by the prior's risk score $\eta$. For each comparable pair $\mathcal{P} = \{(i, j) : \eta_i < \eta_j\}$ (patient $i$ is lower-risk and should therefore have a longer predicted survival time than $j$):

$$\mathcal{L}_{\text{rank}} = -\frac{1}{|\mathcal{P}|} \sum_{(i,j) \in \mathcal{P}} \log \sigma(\hat{\mu}_i - \hat{\mu}_j), \tag{5}$$

where $\sigma(\cdot)$ is the logistic sigmoid. This follows the standard RankNet pairwise loss (Burges et al., 2005).

**Combined objectives**   We train four total variants: `SurvPFN [N]`, `SurvPFN [NR]`, `SurvPFN [O]`, and `SurvPFN [OR]`, where the `+R` variants add the ranking term to the corresponding base loss with equal weighting. Per-dataset results for all four are reported in Table 1.

## B.4. Artificial Intelligence Statement

Large language models were used in this work: Anthropic's Claude Code (Opus 4.6 and 4.7) assisted with code implementation, while Anthropic's Claude and Mistral's Le Chat supported writing tasks.

*Table 1.* Per-dataset performance of all evaluated models. Dataset rows are annotated with $(n, n_{cat}, n_{num})$ giving the sample count and the number of categorical / numeric features (each categorical column counted once, not once per OHE indicator). Values are mean$_{\pm std}$ across cross-validation folds, with leading zeros omitted. **Bold** marks the best model per dataset. Higher is better for C-index (↑); lower is better for IBS (↓). † denotes ablation variants.

| | Baselines | | | TFMRegression (ours) | | | | Ablations | | | |
|---|---|---|---|---|---|---|---|---|---|---|---|
| Dataset $(n, n_{cat}, n_{num})$ | Cox PH | DeepSurv | RSF | SurvPFN [N] | SurvPFN [NR] | SurvPFN [O] | SurvPFN [OR] | SurvPFN [N]† | SurvPFN [NR]† | SurvPFN [O]† | SurvPFN [OR]† |
| *(a) Concordance Index (↑)* | | | | | | | | | | | |
| Bergamaschi (82, 0, 10) | $.620_{\pm.139}$ | $.594_{\pm.100}$ | $.622_{\pm.119}$ | $.608_{\pm.129}$ | $.604_{\pm.108}$ | $\mathbf{.675_{\pm.133}}$ | $.642_{\pm.094}$ | $.617_{\pm.130}$ | $.638_{\pm.116}$ | $.541_{\pm.120}$ | $.566_{\pm.056}$ |
| breast (100, 4, 0) | $\mathbf{.814_{\pm.045}}$ | $.744_{\pm.119}$ | $.773_{\pm.069}$ | $.807_{\pm.050}$ | $.804_{\pm.049}$ | $.781_{\pm.073}$ | $.809_{\pm.052}$ | $.796_{\pm.070}$ | $.803_{\pm.061}$ | $.807_{\pm.050}$ | $.809_{\pm.052}$ |
| cancer (228, 3, 5) | $.577_{\pm.017}$ | $.579_{\pm.080}$ | $.589_{\pm.052}$ | $.578_{\pm.041}$ | $\mathbf{.608_{\pm.024}}$ | $.560_{\pm.036}$ | $.582_{\pm.013}$ | $.563_{\pm.043}$ | $.561_{\pm.037}$ | $.579_{\pm.032}$ | $.579_{\pm.057}$ |
| cgd (128, 7, 3) | $.528_{\pm.030}$ | $.421_{\pm.185}$ | $.415_{\pm.176}$ | $.560_{\pm.141}$ | $.444_{\pm.194}$ | $.377_{\pm.190}$ | $.550_{\pm.088}$ | $\mathbf{.661_{\pm.226}}$ | $.500_{\pm.148}$ | $.580_{\pm.086}$ | $.646_{\pm.206}$ |
| colon (929, 7, 2) | $.655_{\pm.020}$ | $.622_{\pm.043}$ | $.659_{\pm.022}$ | $.663_{\pm.016}$ | $.653_{\pm.027}$ | $.652_{\pm.019}$ | $.654_{\pm.020}$ | $\mathbf{.663_{\pm.018}}$ | $.663_{\pm.016}$ | $.647_{\pm.025}$ | $.652_{\pm.023}$ |
| diabetes (394, 3, 1) | $\mathbf{.607_{\pm.075}}$ | $.605_{\pm.055}$ | $.580_{\pm.023}$ | $.587_{\pm.068}$ | $.586_{\pm.052}$ | $.555_{\pm.109}$ | $.596_{\pm.060}$ | $.562_{\pm.059}$ | $.581_{\pm.046}$ | $.547_{\pm.033}$ | $.569_{\pm.048}$ |
| e1684 (284, 2, 1) | $.559_{\pm.044}$ | $.517_{\pm.018}$ | $.532_{\pm.012}$ | $.551_{\pm.036}$ | $\mathbf{.564_{\pm.067}}$ | $.551_{\pm.065}$ | $.542_{\pm.038}$ | $.561_{\pm.040}$ | $.557_{\pm.041}$ | $.553_{\pm.048}$ | $.562_{\pm.051}$ |
| follic (541, 3, 2) | $\mathbf{.629_{\pm.036}}$ | $.594_{\pm.046}$ | $.599_{\pm.027}$ | $.615_{\pm.029}$ | $.617_{\pm.029}$ | $.621_{\pm.027}$ | $.621_{\pm.033}$ | $.593_{\pm.025}$ | $.608_{\pm.031}$ | $.616_{\pm.026}$ | $.610_{\pm.028}$ |
| GBSG2 (686, 3, 5) | $.676_{\pm.035}$ | $.670_{\pm.027}$ | $.691_{\pm.029}$ | $.694_{\pm.023}$ | $\mathbf{.698_{\pm.021}}$ | $.685_{\pm.032}$ | $.691_{\pm.020}$ | $.633_{\pm.037}$ | $.678_{\pm.029}$ | $.669_{\pm.027}$ | $.684_{\pm.025}$ |
| glioma (37, 3, 1) | $.785_{\pm.086}$ | $.816_{\pm.049}$ | $.826_{\pm.043}$ | $.837_{\pm.077}$ | $.811_{\pm.109}$ | $.806_{\pm.076}$ | $.826_{\pm.135}$ | $\mathbf{.853_{\pm.067}}$ | $.810_{\pm.066}$ | $.803_{\pm.051}$ | $.800_{\pm.065}$ |
| grace (1000, 2, 3) | $.693_{\pm.023}$ | $.691_{\pm.023}$ | $.682_{\pm.021}$ | $\mathbf{.695_{\pm.047}}$ | $.692_{\pm.018}$ | $.686_{\pm.047}$ | $.694_{\pm.026}$ | $.680_{\pm.038}$ | $.668_{\pm.053}$ | $.673_{\pm.024}$ | $.657_{\pm.026}$ |
| Melanoma (205, 2, 3) | $.715_{\pm.066}$ | $.687_{\pm.063}$ | $.714_{\pm.058}$ | $\mathbf{.727_{\pm.078}}$ | $.725_{\pm.070}$ | $.623_{\pm.114}$ | $.718_{\pm.074}$ | $.592_{\pm.118}$ | $.625_{\pm.122}$ | $.715_{\pm.083}$ | $.674_{\pm.099}$ |
| mgus (241, 2, 7) | $.702_{\pm.044}$ | $.685_{\pm.050}$ | $.706_{\pm.046}$ | $.698_{\pm.054}$ | $.694_{\pm.052}$ | $.686_{\pm.039}$ | $.700_{\pm.050}$ | $.700_{\pm.044}$ | $\mathbf{.709_{\pm.043}}$ | $.703_{\pm.040}$ | $.701_{\pm.047}$ |
| ova (358, 3, 2) | $.631_{\pm.039}$ | $.619_{\pm.042}$ | $.647_{\pm.040}$ | $.650_{\pm.042}$ | $.644_{\pm.032}$ | $\mathbf{.652_{\pm.041}}$ | $.648_{\pm.039}$ | $.636_{\pm.052}$ | $.639_{\pm.047}$ | $.641_{\pm.051}$ | $.640_{\pm.053}$ |
| ovarian (26, 3, 1) | $.688_{\pm.290}$ | $.630_{\pm.342}$ | $.637_{\pm.251}$ | $.716_{\pm.283}$ | $.716_{\pm.283}$ | $.716_{\pm.283}$ | $.697_{\pm.261}$ | $.692_{\pm.281}$ | $.688_{\pm.290}$ | $\mathbf{.740_{\pm.292}}$ | $.688_{\pm.290}$ |
| pbc (312, 5, 1) | $\mathbf{.738_{\pm.070}}$ | $.725_{\pm.048}$ | $.717_{\pm.030}$ | $.736_{\pm.068}$ | $.727_{\pm.066}$ | $.708_{\pm.070}$ | $.735_{\pm.067}$ | $.696_{\pm.090}$ | $.729_{\pm.076}$ | $.724_{\pm.062}$ | $.706_{\pm.057}$ |
| retinopathy (394, 5, 2) | $\mathbf{.652_{\pm.044}}$ | $.633_{\pm.048}$ | $.630_{\pm.033}$ | $.627_{\pm.033}$ | $.629_{\pm.056}$ | $.629_{\pm.049}$ | $.642_{\pm.055}$ | $.609_{\pm.057}$ | $.625_{\pm.035}$ | $.602_{\pm.047}$ | $.616_{\pm.051}$ |
| stagec (146, 4, 3) | $.664_{\pm.084}$ | $.586_{\pm.127}$ | $.645_{\pm.092}$ | $.669_{\pm.058}$ | $.667_{\pm.078}$ | $.650_{\pm.119}$ | $.668_{\pm.083}$ | $.656_{\pm.111}$ | $.639_{\pm.115}$ | $\mathbf{.672_{\pm.105}}$ | $.668_{\pm.117}$ |
| uis (628, 5, 3) | $\mathbf{.591_{\pm.010}}$ | $.567_{\pm.007}$ | $.586_{\pm.025}$ | $.588_{\pm.025}$ | $.572_{\pm.042}$ | $.583_{\pm.024}$ | $.572_{\pm.035}$ | $.573_{\pm.027}$ | $.578_{\pm.014}$ | $.585_{\pm.034}$ | $.588_{\pm.031}$ |
| Unemployment (452, 5, 0) | $\mathbf{.556_{\pm.061}}$ | $.500_{\pm.062}$ | $.525_{\pm.066}$ | $.541_{\pm.064}$ | $.543_{\pm.064}$ | $.546_{\pm.043}$ | $.556_{\pm.077}$ | $.548_{\pm.047}$ | $.549_{\pm.054}$ | $.546_{\pm.057}$ | $.552_{\pm.055}$ |
| veteran (137, 3, 3) | $.693_{\pm.046}$ | $.648_{\pm.088}$ | $\mathbf{.719_{\pm.039}}$ | $.695_{\pm.026}$ | $.692_{\pm.025}$ | $.708_{\pm.056}$ | $.699_{\pm.030}$ | $.693_{\pm.014}$ | $.705_{\pm.037}$ | $.710_{\pm.052}$ | $.711_{\pm.045}$ |
| Z243 (100, 5, 4) | $.908_{\pm.041}$ | $.874_{\pm.063}$ | $.853_{\pm.014}$ | $.918_{\pm.025}$ | $.930_{\pm.024}$ | $.908_{\pm.034}$ | $\mathbf{.942_{\pm.015}}$ | $.914_{\pm.033}$ | $.924_{\pm.024}$ | $.911_{\pm.019}$ | $.918_{\pm.023}$ |
| *Mean* | .667 | .637 | .652 | .671 | .665 | .653 | **.672** | .659 | .658 | .662 | .664 |
| *(b) Integrated Brier Score (↓)* | | | | | | | | | | | |
| Bergamaschi (82, 0, 10) | $\mathbf{.189_{\pm.044}}$ | $.255_{\pm.115}$ | $.205_{\pm.069}$ | $.234_{\pm.096}$ | $.245_{\pm.104}$ | $.234_{\pm.111}$ | $.236_{\pm.111}$ | $.407_{\pm.314}$ | $.374_{\pm.278}$ | $.336_{\pm.246}$ | $.299_{\pm.191}$ |
| breast (100, 4, 0) | $.108_{\pm.013}$ | $.114_{\pm.015}$ | $.114_{\pm.018}$ | $\mathbf{.106_{\pm.010}}$ | $.112_{\pm.014}$ | $.116_{\pm.014}$ | $.109_{\pm.013}$ | $.149_{\pm.017}$ | $.171_{\pm.026}$ | $.111_{\pm.017}$ | $.109_{\pm.014}$ |
| cancer (228, 3, 5) | $.175_{\pm.018}$ | $\mathbf{.171_{\pm.023}}$ | $.171_{\pm.019}$ | $.177_{\pm.014}$ | $.178_{\pm.007}$ | $.188_{\pm.012}$ | $.173_{\pm.014}$ | $.175_{\pm.013}$ | $.171_{\pm.012}$ | $.414_{\pm.030}$ | $.410_{\pm.021}$ |
| cgd (128, 7, 3) | $.237_{\pm.053}$ | $.263_{\pm.124}$ | $.193_{\pm.052}$ | $.186_{\pm.037}$ | $.185_{\pm.040}$ | $.184_{\pm.031}$ | $\mathbf{.181_{\pm.031}}$ | $.216_{\pm.047}$ | $.208_{\pm.049}$ | $.194_{\pm.029}$ | $.196_{\pm.033}$ |
| colon (929, 7, 2) | $.189_{\pm.011}$ | $.211_{\pm.040}$ | $\mathbf{.185_{\pm.013}}$ | $.189_{\pm.011}$ | $.188_{\pm.012}$ | $.191_{\pm.010}$ | $.188_{\pm.012}$ | $.213_{\pm.014}$ | $.221_{\pm.018}$ | $.203_{\pm.008}$ | $.194_{\pm.009}$ |
| diabetes (394, 3, 1) | $.206_{\pm.016}$ | $\mathbf{.203_{\pm.015}}$ | $.217_{\pm.019}$ | $.208_{\pm.014}$ | $.208_{\pm.016}$ | $.210_{\pm.011}$ | $.211_{\pm.017}$ | $.241_{\pm.036}$ | $.260_{\pm.038}$ | $.213_{\pm.011}$ | $.209_{\pm.011}$ |
| e1684 (284, 2, 1) | $\mathbf{.233_{\pm.031}}$ | $.248_{\pm.027}$ | $.258_{\pm.032}$ | $.257_{\pm.046}$ | $.255_{\pm.054}$ | $.249_{\pm.048}$ | $.251_{\pm.053}$ | $.261_{\pm.055}$ | $.265_{\pm.058}$ | $.238_{\pm.038}$ | $.239_{\pm.036}$ |
| follic (541, 3, 2) | $.191_{\pm.034}$ | $.201_{\pm.033}$ | $.195_{\pm.020}$ | $.191_{\pm.028}$ | $\mathbf{.185_{\pm.025}}$ | $.191_{\pm.029}$ | $.185_{\pm.024}$ | $.204_{\pm.035}$ | $.201_{\pm.034}$ | $.200_{\pm.033}$ | $.192_{\pm.030}$ |
| GBSG2 (686, 3, 5) | $.176_{\pm.021}$ | $.188_{\pm.034}$ | $.170_{\pm.016}$ | $.175_{\pm.014}$ | $\mathbf{.169_{\pm.016}}$ | $.189_{\pm.024}$ | $.177_{\pm.017}$ | $.214_{\pm.022}$ | $.201_{\pm.023}$ | $.230_{\pm.019}$ | $.213_{\pm.016}$ |
| glioma (37, 3, 1) | $.133_{\pm.020}$ | $.147_{\pm.021}$ | $.140_{\pm.025}$ | $.137_{\pm.034}$ | $\mathbf{.124_{\pm.030}}$ | $.138_{\pm.039}$ | $.130_{\pm.042}$ | $.142_{\pm.037}$ | $.140_{\pm.040}$ | $.144_{\pm.040}$ | $.153_{\pm.053}$ |
| grace (1000, 2, 3) | $\mathbf{.173_{\pm.004}}$ | $.176_{\pm.006}$ | $.176_{\pm.010}$ | $.175_{\pm.012}$ | $.177_{\pm.008}$ | $.190_{\pm.012}$ | $.178_{\pm.010}$ | $.234_{\pm.013}$ | $.255_{\pm.019}$ | $.184_{\pm.008}$ | $.188_{\pm.006}$ |
| Melanoma (205, 2, 3) | $.170_{\pm.042}$ | $.166_{\pm.033}$ | $.174_{\pm.044}$ | $.154_{\pm.031}$ | $.153_{\pm.031}$ | $\mathbf{.138_{\pm.018}}$ | $.153_{\pm.032}$ | $.203_{\pm.044}$ | $.180_{\pm.036}$ | $.166_{\pm.021}$ | $.148_{\pm.018}$ |
| mgus (241, 2, 7) | $\mathbf{.126_{\pm.011}}$ | $.152_{\pm.047}$ | $.129_{\pm.009}$ | $.149_{\pm.012}$ | $.147_{\pm.014}$ | $.155_{\pm.012}$ | $.148_{\pm.011}$ | $.166_{\pm.007}$ | $.149_{\pm.009}$ | $.368_{\pm.048}$ | $.295_{\pm.038}$ |
| ova (358, 3, 2) | $.190_{\pm.018}$ | $.196_{\pm.018}$ | $.197_{\pm.017}$ | $.189_{\pm.016}$ | $.190_{\pm.017}$ | $.195_{\pm.017}$ | $\mathbf{.189_{\pm.017}}$ | $.194_{\pm.013}$ | $.189_{\pm.017}$ | $.337_{\pm.031}$ | $.318_{\pm.037}$ |
| ovarian (26, 3, 1) | $.300_{\pm.273}$ | $.458_{\pm.273}$ | $.264_{\pm.131}$ | $.239_{\pm.104}$ | $\mathbf{.232_{\pm.159}}$ | $.254_{\pm.154}$ | $.280_{\pm.174}$ | $.306_{\pm.205}$ | $.304_{\pm.196}$ | $.260_{\pm.132}$ | $.254_{\pm.113}$ |
| pbc (312, 5, 1) | $.161_{\pm.024}$ | $.164_{\pm.026}$ | $.167_{\pm.018}$ | $.162_{\pm.018}$ | $\mathbf{.159_{\pm.017}}$ | $.173_{\pm.023}$ | $.162_{\pm.018}$ | $.204_{\pm.031}$ | $.215_{\pm.042}$ | $.241_{\pm.010}$ | $.200_{\pm.018}$ |
| retinopathy (394, 5, 2) | $.190_{\pm.020}$ | $.202_{\pm.026}$ | $.196_{\pm.017}$ | $.195_{\pm.018}$ | $.195_{\pm.017}$ | $.198_{\pm.019}$ | $.196_{\pm.017}$ | $.239_{\pm.040}$ | $.244_{\pm.039}$ | $.203_{\pm.013}$ | $.199_{\pm.014}$ |
| stagec (146, 4, 3) | $.243_{\pm.100}$ | $.272_{\pm.095}$ | $.235_{\pm.086}$ | $.234_{\pm.096}$ | $.241_{\pm.116}$ | $.214_{\pm.093}$ | $.238_{\pm.114}$ | $.313_{\pm.130}$ | $.319_{\pm.138}$ | $.217_{\pm.068}$ | $.230_{\pm.071}$ |
| uis (628, 5, 3) | $.181_{\pm.003}$ | $\mathbf{.180_{\pm.003}}$ | $.181_{\pm.009}$ | $.181_{\pm.006}$ | $.182_{\pm.007}$ | $.185_{\pm.006}$ | $.182_{\pm.006}$ | $.186_{\pm.007}$ | $.183_{\pm.006}$ | $.350_{\pm.020}$ | $.302_{\pm.015}$ |
| Unemployment (452, 5, 0) | $.192_{\pm.012}$ | $.205_{\pm.013}$ | $.195_{\pm.016}$ | $\mathbf{.189_{\pm.005}}$ | $.191_{\pm.009}$ | $.206_{\pm.019}$ | $.195_{\pm.013}$ | $.209_{\pm.015}$ | $.219_{\pm.017}$ | $.194_{\pm.005}$ | $.193_{\pm.007}$ |
| veteran (137, 3, 3) | $.136_{\pm.009}$ | $.151_{\pm.019}$ | $\mathbf{.133_{\pm.005}}$ | $.139_{\pm.011}$ | $.143_{\pm.016}$ | $.142_{\pm.015}$ | $.138_{\pm.013}$ | $.143_{\pm.007}$ | $.143_{\pm.016}$ | $.187_{\pm.050}$ | $.218_{\pm.062}$ |
| Z243 (100, 5, 4) | $\mathbf{.048_{\pm.013}}$ | $.071_{\pm.019}$ | $.076_{\pm.017}$ | $.068_{\pm.018}$ | $.059_{\pm.008}$ | $.055_{\pm.008}$ | $.057_{\pm.008}$ | $.059_{\pm.011}$ | $.059_{\pm.009}$ | $.139_{\pm.022}$ | $.128_{\pm.023}$ |
| *Mean* | .179 | .200 | .181 | .179 | **.178** | .182 | .180 | .213 | .212 | .233 | .222 |
| *(c) Integrated Calibration Index (↓)* | | | | | | | | | | | |
| Bergamaschi (82, 0, 10) | $.143_{\pm.058}$ | $.183_{\pm.035}$ | $.162_{\pm.058}$ | $\mathbf{.119_{\pm.072}}$ | $.156_{\pm.043}$ | $.141_{\pm.042}$ | $.128_{\pm.046}$ | $.284_{\pm.044}$ | $.227_{\pm.067}$ | $.211_{\pm.047}$ | $.256_{\pm.065}$ |
| breast (100, 4, 0) | $\mathbf{.097_{\pm.051}}$ | $.120_{\pm.044}$ | $.128_{\pm.079}$ | $.115_{\pm.011}$ | $.106_{\pm.045}$ | $.121_{\pm.018}$ | $.098_{\pm.040}$ | $.318_{\pm.034}$ | $.407_{\pm.031}$ | $.144_{\pm.009}$ | $.125_{\pm.032}$ |
| cancer (228, 3, 5) | $.103_{\pm.038}$ | $.086_{\pm.013}$ | $.110_{\pm.040}$ | $.089_{\pm.020}$ | $.086_{\pm.022}$ | $.086_{\pm.036}$ | $.086_{\pm.013}$ | $.099_{\pm.038}$ | $\mathbf{.078_{\pm.038}}$ | $.278_{\pm.043}$ | $.278_{\pm.058}$ |
| cgd (128, 7, 3) | $.215_{\pm.074}$ | $.199_{\pm.069}$ | $.129_{\pm.024}$ | $.140_{\pm.085}$ | $\mathbf{.106_{\pm.057}}$ | $.142_{\pm.066}$ | $.120_{\pm.062}$ | $.233_{\pm.040}$ | $.238_{\pm.040}$ | $.147_{\pm.025}$ | $.158_{\pm.043}$ |
| colon (929, 7, 2) | $.080_{\pm.091}$ | $.044_{\pm.012}$ | $\mathbf{.038_{\pm.021}}$ | $.063_{\pm.014}$ | $.050_{\pm.013}$ | $.055_{\pm.018}$ | $.062_{\pm.020}$ | $.158_{\pm.023}$ | $.184_{\pm.023}$ | $.115_{\pm.030}$ | $.089_{\pm.033}$ |
| diabetes (394, 3, 1) | $.084_{\pm.015}$ | $.079_{\pm.014}$ | $.113_{\pm.044}$ | $.090_{\pm.022}$ | $.085_{\pm.022}$ | $.096_{\pm.017}$ | $.096_{\pm.017}$ | $.165_{\pm.025}$ | $.228_{\pm.028}$ | $.089_{\pm.030}$ | $\mathbf{.065_{\pm.028}}$ |
| e1684 (284, 2, 1) | $.080_{\pm.017}$ | $.107_{\pm.053}$ | $.146_{\pm.031}$ | $.104_{\pm.018}$ | $.092_{\pm.038}$ | $.090_{\pm.031}$ | $\mathbf{.069_{\pm.019}}$ | $.083_{\pm.030}$ | $.076_{\pm.028}$ | $.096_{\pm.040}$ | $.084_{\pm.017}$ |
| follic (541, 3, 2) | $.051_{\pm.021}$ | $\mathbf{.045_{\pm.022}}$ | $.057_{\pm.019}$ | $.061_{\pm.020}$ | $.048_{\pm.026}$ | $.075_{\pm.032}$ | $.050_{\pm.034}$ | $.083_{\pm.024}$ | $.090_{\pm.030}$ | $.079_{\pm.022}$ | $.065_{\pm.039}$ |
| GBSG2 (686, 3, 5) | $.062_{\pm.021}$ | $.072_{\pm.061}$ | $\mathbf{.050_{\pm.016}}$ | $.071_{\pm.020}$ | $.053_{\pm.014}$ | $.087_{\pm.019}$ | $.075_{\pm.019}$ | $.172_{\pm.034}$ | $.156_{\pm.029}$ | $.177_{\pm.033}$ | $.154_{\pm.035}$ |
| glioma (37, 3, 1) | $\mathbf{.105_{\pm.045}}$ | $.183_{\pm.072}$ | $.144_{\pm.031}$ | $.159_{\pm.060}$ | $.127_{\pm.026}$ | $.151_{\pm.056}$ | $.138_{\pm.052}$ | $.121_{\pm.028}$ | $.147_{\pm.035}$ | $.145_{\pm.034}$ | $.143_{\pm.084}$ |
| grace (1000, 2, 3) | $\mathbf{.049_{\pm.012}}$ | $.082_{\pm.058}$ | $.062_{\pm.035}$ | $.076_{\pm.026}$ | $.056_{\pm.016}$ | $.133_{\pm.015}$ | $.077_{\pm.018}$ | $.303_{\pm.020}$ | $.349_{\pm.020}$ | $.081_{\pm.033}$ | $.092_{\pm.019}$ |
| Melanoma (205, 2, 3) | $.118_{\pm.039}$ | $\mathbf{.096_{\pm.033}}$ | $.119_{\pm.038}$ | $.108_{\pm.027}$ | $.112_{\pm.032}$ | $.124_{\pm.031}$ | $.110_{\pm.025}$ | $.263_{\pm.029}$ | $.274_{\pm.038}$ | $.125_{\pm.044}$ | $.097_{\pm.032}$ |
| mgus (241, 2, 7) | $\mathbf{.058_{\pm.020}}$ | $.096_{\pm.031}$ | $.058_{\pm.012}$ | $.116_{\pm.039}$ | $.104_{\pm.040}$ | $.092_{\pm.024}$ | $.119_{\pm.044}$ | $.116_{\pm.044}$ | $.089_{\pm.034}$ | $.342_{\pm.051}$ | $.262_{\pm.032}$ |
| ova (358, 3, 2) | $.068_{\pm.024}$ | $\mathbf{.064_{\pm.010}}$ | $.085_{\pm.013}$ | $.075_{\pm.037}$ | $.069_{\pm.030}$ | $.116_{\pm.045}$ | $.074_{\pm.038}$ | $.078_{\pm.048}$ | $.075_{\pm.029}$ | $.318_{\pm.048}$ | $.290_{\pm.056}$ |
| ovarian (26, 3, 1) | $.400_{\pm.233}$ | $.350_{\pm.057}$ | $.300_{\pm.100}$ | $.289_{\pm.105}$ | $\mathbf{.283_{\pm.119}}$ | $.338_{\pm.157}$ | $.300_{\pm.114}$ | $.308_{\pm.108}$ | $.340_{\pm.117}$ | $.306_{\pm.164}$ | $.337_{\pm.181}$ |
| pbc (312, 5, 1) | $.099_{\pm.050}$ | $\mathbf{.051_{\pm.027}}$ | $.061_{\pm.035}$ | $.095_{\pm.032}$ | $.075_{\pm.035}$ | $.122_{\pm.038}$ | $.096_{\pm.032}$ | $.206_{\pm.044}$ | $.247_{\pm.034}$ | $.178_{\pm.034}$ | $.112_{\pm.032}$ |
| retinopathy (394, 5, 2) | $\mathbf{.049_{\pm.013}}$ | $.087_{\pm.060}$ | $.082_{\pm.032}$ | $.080_{\pm.024}$ | $.087_{\pm.025}$ | $.094_{\pm.029}$ | $.098_{\pm.019}$ | $.180_{\pm.031}$ | $.218_{\pm.028}$ | $.090_{\pm.037}$ | $.073_{\pm.039}$ |
| stagec (146, 4, 3) | $.134_{\pm.044}$ | $.203_{\pm.086}$ | $.138_{\pm.055}$ | $.139_{\pm.052}$ | $.151_{\pm.043}$ | $\mathbf{.122_{\pm.047}}$ | $.148_{\pm.049}$ | $.163_{\pm.060}$ | $.144_{\pm.076}$ | $.145_{\pm.067}$ | $.126_{\pm.071}$ |
| uis (628, 5, 3) | $.060_{\pm.022}$ | $.069_{\pm.030}$ | $.062_{\pm.022}$ | $.073_{\pm.022}$ | $.072_{\pm.038}$ | $.069_{\pm.030}$ | $.070_{\pm.032}$ | $.060_{\pm.014}$ | $\mathbf{.059_{\pm.017}}$ | $.297_{\pm.046}$ | $.252_{\pm.049}$ |
| Unemployment (452, 5, 0) | $.075_{\pm.042}$ | $.092_{\pm.037}$ | $.087_{\pm.024}$ | $\mathbf{.067_{\pm.017}}$ | $.076_{\pm.030}$ | $.109_{\pm.037}$ | $.094_{\pm.032}$ | $.143_{\pm.043}$ | $.179_{\pm.040}$ | $.100_{\pm.056}$ | $.111_{\pm.052}$ |
| veteran (137, 3, 3) | $\mathbf{.111_{\pm.020}}$ | $.126_{\pm.040}$ | $.155_{\pm.051}$ | $.142_{\pm.035}$ | $.121_{\pm.049}$ | $.145_{\pm.064}$ | $.151_{\pm.044}$ | $.128_{\pm.027}$ | $.129_{\pm.050}$ | $.152_{\pm.054}$ | $.167_{\pm.056}$ |
| Z243 (100, 5, 4) | $.046_{\pm.024}$ | $.144_{\pm.083}$ | $.112_{\pm.051}$ | $.143_{\pm.034}$ | $.106_{\pm.041}$ | $.107_{\pm.041}$ | $.084_{\pm.026}$ | $.140_{\pm.029}$ | $.093_{\pm.028}$ | $.044_{\pm.026}$ | $\mathbf{.033_{\pm.015}}$ |
| *Mean* | .104 | .117 | .109 | .110 | **.101** | .119 | .107 | .173 | .183 | .166 | .153 |

*Table 2.* Per-metric average ranks across all 22 SurvSet datasets (lower is better). We report ranks separately for the concordance index (C-Index), the Integrated Brier Score (IBS), and the Integrated Calibration Index (ICI). `SurvPFN` variants are denoted by their training configuration: `[N]` native censoring loss, `[O]` oracle targets, `[R]` auxiliary ranking loss, and combinations thereof. Ablation models (`abl.`) force the event indicator feature to 1. **Bold** marks the model highlighted in the main text.

| Model | Rank C Score | Model | Rank IBS | Model | Rank ICI |
|---|---|---|---|---|---|
| SurvPFN [OR] | 4,25 | Cox PH | 3,77 | Cox PH | 4,05 |
| SurvPFN [N] | 4,84 | **SurvPFN [NR]** | **4,00** | **SurvPFN [NR]** | **4,09** |
| Cox PH | 5,27 | SurvPFN [N] | 4,09 | SurvPFN [OR] | 5,41 |
| BinSurv [MITRA] | 5,73 | SurvPFN [OR] | 4,45 | SurvPFN [N] | 5,68 |
| **SurvPFN [NR]** | **5,77** | RSF | 5,73 | RSF | 5,77 |
| SurvPFN [OR] abl. | 6,66 | SurvPFN [O] | 6,05 | DeepSurv | 5,95 |
| SurvPFN [O] abl. | 6,80 | BinSurv [MITRA] | 7,27 | BinSurv [MITRA] | 6,86 |
| SurvPFN [NR] abl. | 6,82 | DeepSurv | 7,41 | SurvPFN [O] | 6,91 |
| RSF | 7,18 | SurvPFN [OR] abl. | 7,59 | SurvPFN [OR] abl. | 7,32 |
| SurvPFN [O] | 7,50 | SurvPFN [O] abl. | 8,64 | SurvPFN [NR] abl. | 8,50 |
| SurvPFN [N] abl. | 7,77 | SurvPFN [NR] abl. | 9,27 | SurvPFN [O] abl. | 8,68 |
| DeepSurv | 9,41 | SurvPFN [N] abl. | 9,73 | SurvPFN [N] abl. | 8,77 |

