# OpenReview forum: "SurvPFN: Towards Foundation Models for Survival Predictions"
_ICML.cc/2026/Workshop/FMSD — FMSD @ ICML 2026 Poster_

### Official Review · Reviewer_bZG4 · 2026-05-13

**Rating:** 7
**Confidence:** 4

**Review:**

**Summary**: This work proposes Prior-data Fitted Networks (PFNs) for survival regression, involving training a small regression PFN from scratch on a survival prior.

**Strengths**:
- The method involves pretraining the model from scratch using curated synthetic survival training data, as opposed to merely applying a PFN off-the-shelf. Both the modeling approach and the synthetic data generation are contributions.

- The synthetic survival training data includes scenarios of proportional hazards (PH) and non-proportional hazards, reflecting more realistic survival curves compared to PH scenarios alone.

- The approach supports censored data within the modeling framework itself, as opposed to using imputation or discarding censored observations.

- The approach is evaluated on 22 real-world datasets from SurvSet.

- Evaluation spans several metrics, including the C-index, Integrated Brier Score, and Integrated Calibration Index.

**Areas for Improvement**:

- In standard survival models, the event indicator is part of the outcome tuple and enters only through the loss function or splitting criterion; it is never included as a covariate in X, as doing so would leak target information. The authors should clearly explain how the event indicator is presented to the model using both text and mathematical notation. PFNs operate differently in that context rows are analogous to training data shown in-context, which changes how the event indicator can legitimately be incorporated. Explicitly clarifying this distinction would help prevent the impression that target information is being leaked.

- A discussion outlining the reasons for selecting the Weibull distribution, beyond noting that it is commonly used in survival analysis, would better motivate this design choice for readers.

- Results should be reported quantitatively, not only qualitatively. Rather than stating the method is "competitive with" baselines, the authors should report percentage improvements or statistically test whether prediction error significantly degrades or improves relative to each baseline. The comparable results in terms of statistical significance is noted in the results section but not in the Abstract.

- The descriptions of survival time generation and censoring mechanisms should be formalized mathematically. The paper describes the methods mostly in text.

- The paper would benefit from more clearly delineating its contributions: (1) a PFN approach based on the Weibull distribution that handles censored data without imputation or exclusion, (2) a synthetic survival training data generation pipeline, and (3) a framework for training a PFN from scratch for survival regression.

- The results appear to fall within error bounds across all models, which is acknowledged in the paper. However, a key advantage of this approach is that the PFN, pretrained on synthetic data, requires no retraining per dataset, unlike the baselines. This could be emphasized. Also, demonstrating the degradation in baseline model performance in zero-shot scenarios would help strengthen this narrative.

- The work compares against four survival regression baselines. Additional models such as Deep Survival Machines (DSM) and Deep Cox Mixtures (DCM) could be included.

**Justification of Score:** Overall, this is an interesting paper, as I have not seen a PFN approach for survival regression that incorporates pretraining on synthetic survival data, as opposed to simply applying a PFN off-the-shelf to regress survival times. Handling right-censored data without imputation or data exclusion is also an important contribution. As such, this work will be relevant to readers interested in Tabular Foundation Models and Survival Analysis. However, I believe the points raised in the Areas for Improvement should be addressed to help clarify and strengthen the work.

---

### Official Review · Reviewer_pDkf · 2026-05-21
**PFN adaptation for survival prediction, but empirical claims could be stronger**

**Rating:** 6
**Confidence:** 4

**Review:**

I sense main idea is to adapt a small NanoTabPFN-style regression model to time-to-event data by training on synthetic Weibull survival tasks, adding censoring information, and optimizing a censored likelihood objective. The model is evaluated on 22 SurvSet datasets and is shown to be competitive with CoxPH, RSF, DeepSurv, and BinSurv, without per-dataset fitting.

Strengths:
The paper tackles an important and underexplored problem: bringing tabular foundation models to survival analysis, where censoring makes standard regression unsuitable. I liked the framing of survival prediction as continuous-time distributional regression with a censored loss. The method is also appealingly simple: it uses a relatively small PFN backbone and avoids task-specific fitting or heavy feature engineering. The evaluation is reasonably broad, covering multiple SurvSet datasets and using several metrics: C-index, IBS, and ICI. The ablation on the event indicator is useful and supports the claim that explicit censoring information matters, especially for calibration.

Areas for Improvement:
The main weakness is that the empirical advantage is still modest. The paper states that no pairwise comparison reaches statistical significance, and CoxPH appears to perform best overall in the main results. This makes the contribution more of a promising proof-of-concept. The evaluation is also limited to small datasets with at most 1000 rows and 10 features. That is understandable given the NanoTabPFN backbone, but it limits the strength of the foundation-model claim. It would be helpful to show examples of  benefit in zero-shot deployment time or robustness to small sample size.

Detailed Comments:
1. The paper would benefit from a clearer explanation of why the event indicator is appended as a feature rather than handled only through the loss.
2. Since CoxPH is strong in the results, the authors should more explicitly state the use case where SurvPFN is preferable.
3. The comparison to BinSurv is useful, but the paper could better explain the tradeoff between continuous-time density prediction and discrete-time binning.

Justification of Score:
Overall, I think this is a solid submission. However, the current results are more competitive than clearly superior. I would lean toward acceptance because the paper opens an interesting direction for foundation models on structured survival data, but I would not rate it as a strong accept yet.